

# Application of carbon dioxide to the skin and muscle oxygenation of human lower-limb muscle sites during cold water immersion

Miho Yoshimura[1,2], Tatsuya Hojo[1], Hayato Yamamoto[1], Misato Tachibana[1], Masatoshi Nakamura[3], Hiroaki Tsutsumi[4,5] and Yoshiyuki Fukuoka[1]

[1] Faculty of Health and Sports Science, Doshisha University, Kyotanabe, Kyoto, Japan
[2] Division of Sports Facility Service, Mizuno Corporation, Osaka, Osaka, Japan
[3] Department of Physical Therapy, Niigata University of Health and Warfare, Niigata, Niigata, Japan
[4] Faculty of Environmental and Symbiotic Science, Prefectural University of Kumamoto, Kumamoto, Kumamoto, Japan
[5] Division of eco-Bubble® development, Taikohgiken Itd., Kumamoto, Kumamoto, Japan

Corresponding author
Yoshiyuki Fukuoka,
yfukuoka@mail.doshisha.ac.jp

## ABSTRACT

**Background**. Cold therapy has the disadvantage of inducing vasoconstriction in arterial and venous capillaries. The effects of carbon dioxide ($CO_2$) hot water depend mainly on not only cutaneous vasodilation but also muscle vasodilation. We examined the effects of artificial $CO_2$ cold water immersion (CCWI) on skin oxygenation and muscle oxygenation and the immersed skin temperature.

**Subjects and Methods**. Fifteen healthy young males participated. $CO_2$-rich water containing $CO_2 > 1,150$ ppm was prepared using a micro-bubble device. Each subject's single leg was immersed up to the knee in the $CO_2$-rich water (20 °C) for 15 min, followed by a 20-min recovery period. As a control study, a leg of the subject was immersed in cold tap-water at 20 °C (CWI). The skin temperature at the lower leg under water immersion ($T_{sk}$-WI) and the subject's thermal sensation at the immersed and non-immersed lower legs were measured throughout the experiment. We simultaneously measured the relative changes of local muscle oxygenation/deoxygenation compared to the basal values ($\Delta$oxy[Hb+Mb], $\Delta$deoxy[Hb+Mb], and $\Delta$total[Hb+Mb]) at rest, which reflected the blood flow in the muscle, and we measured the tissue $O_2$ saturation ($S_tO_2$) by near-infrared spectroscopy on two regions of the tibialis anterior (TA) and gastrocnemius (GAS) muscles.

**Results**. Compared to the CWI results, the $\Delta$oxy[Hb+Mb] and $\Delta$total[Hb+Mb] in the TA muscle at CCWI were increased and continued at a steady state during the recovery period. In GAS muscle, the $\Delta$total[Hb+Mb] and $\Delta$deoxy[Hb+Mb] were increased during CCWI compared to CWI. Notably, $S_tO_2$ values in both TA and GAS muscles were significantly increased during CCWI compared to CWI. In addition, compared to the CWI, a significant decrease in $T_{sk}$ at the immersed leg after the CCWI was maintained until the end of the 20-min recovery, and the significant reduction continued.

**Discussion**. The combination of $CO_2$ and cold water can induce both more increased blood inflow into muscles and volume-related (total heme concentration) changes in

deoxy[Hb+Mb] during the recovery period. The $T_{sk}$-WI stayed lower with the CCWI compared to the CWI, as it is associated with vasodilation by $CO_2$.

## INTRODUCTION

Cold therapy has the disadvantage of inducing vasoconstriction in arterial and venous capillaries. The protocols used for cold therapy have involved various combinations of temperatures and durations that were selected based on experience rather than scientific evidence. A review of published protocols indicates that lower water temperature ($\leq 15\,°C$) for durations of $\leq 5$ min have induced many changes in physiological parameters (*Šrámek et al., 2000*; *Bleakley & Davison, 2010*; *Ihsan, Watson & Abbiss, 2016*). The physiological mechanisms by which cold water immersion (CWI) influences the body's recovery are not entirely clear (*White & Wells, 2013*), but these mechanisms are likely to be related to effects of the removal of body heat, reduced muscle temperature, and hydrostatic pressure effects rather than to the cold shock response.

We have questioned whether CWI therapy requires a water temperature $\leq 15\,°C$. Similar reductions in femoral artery blood flow were observed in subjects' responses to $8\,°C$ and $22\,°C$ cooling at rest (*Gregson et al., 2011*) and post-exercise (*Mawhinney et al., 2013*) despite remarkable differences in the muscle temperature. We thus selected the relatively higher temperature of $20\,°C$ for the present study, because a less-noxious cooling temperature that does not cause cold pain (i.e., $>18\,°C$) (*Wolf & Hardy, 1941*) may provide a suitable alterative for individuals who are unable to tolerate colder temperatures (*Gregson et al., 2011*). In addition, *Proulx, Ducharme & Kenny (2003)* observed an increase in shivering rates in previously hyperthermic individuals when CWI at $14\,°C$ was continued beyond 10 min. Indeed, involuntary muscle contraction associated with shivering can increase the body's metabolic rate (*Stocks et al., 2004*).

Over the last three decades, researchers observed that the main effects of immersion in hot water with dissolved carbon dioxide ($CO_2$) are cutaneous vasodilation and muscle vasodilation. These effects are elicited by the diffusion of $CO_2$ through the skin layers into the subcutaneous tissues (*Schnizer et al., 1985*; Komoto et al., 1986; *Ito, Moore & Koss, 1989*; *Hartmann, Bassenge & Pittler, 1997*). We hypothesized that the combined CWI protocol of the relative higher temperature of $20\,°C$ and longer duration could make it possible for $CO_2$ to overcome the vasoconstriction in arterial and venous capillaries that follows a cold stimulus.

Laser Doppler is usually selected for the measurement of the subcutaneous blood (*Schnizer et al., 1985*; *Ito, Moore & Koss, 1989*; *Hartmann, Bassenge & Pittler, 1997*), but near-infrared spectroscopy (NIRS) provides a significant amount of precise information on the oxygenation status within small blood vessels and myocytes, including the status of oxygenation (oxy[Hb+Mb]), deoxygenation (deoxy[Hb+Mb]) and their sum, i.e., the

total heme (total[Hb+Mb]). NIRS has been used to investigate the blood volume-related changes in the oxy[Hb+Mb], (deoxy[Hb+Mb]), and the total[Hb+Mb] concentration profiles (*Adami et al., 2015*; *Binzoni et al., 2000*; *Truijen et al., 2012*) and/or microvascular $O_2$ extraction (*Davies et al., 2008*; *De Roia et al., 2012*; *Ferrari, Muthalib & Quaresima, 2011*). NIRS could thus be useful to determine the effects of $CO_2$ on the changes of microvascular blood flow into skeletal muscles and simultaneously the local metabolism from measurements of the tissue $O_2$ saturation ($S_tO_2$) and deoxy[Hb+Mb] profile throughout cold immersion and recovery phases.

In addition, a customized NIRS device with two additional detector probes can assess the peripheral blood flow in dissociable skin and muscle layers of the tibialis anterior muscle, and its reliability has been validated in both healthy subjects and patients (*Ando et al., 2013*; *Horiuchi et al., 2016*; *Yamabata et al., 2016*). We have therefore attempted to quantify both the skin and muscle blood flow by using this NIRS device in experiments with cold water immersion with enriched $CO_2$ (CCWI). However, it has been unclear whether combined $CO_2$ and cold water would influence both skin and muscle blood perfusion, due to the complexity of various physiological actions (e.g., $CO_2$-induced vasodilation, a cold stimulus, and hydrostatic pressure). Thus, our first hypothesis was that the use of $CO_2$ can overcome the vasoconstriction in arterial and venous capillaries due to a cold stimulus at 20 °C and that consequently, the augmented heat transfer from the body to the water using $CO_2$ could lead to decreased skin temperature during CCWI compared to CWI.

Our second hypothesis was that the arterial and venous vasodilation in both superficial and muscle layers of different muscles would be significantly augmented by enriched $CO_2$, even though during CWI a cold stimulus induces vasoconstriction. We conducted the present study to test these two hypotheses.

## SUBJECTS AND METHODS

### Subjects' characteristics
We recruited 15 healthy young male subjects (age 23.1 ± 0.4 yrs, weight 61.4 ± 1.8 kg, height 171 ± 1.4 cm, mean ± SE) for the study. The subjects' body mass index (BMI) values and body fat percentage were 20.8 ± 0.5 and 15.7 ± 1.1%, respectively. Written informed consent was obtained from all subjects after they received a detailed explanation about all procedures, the purpose of the study, and the possible risks and benefits of their participation. This study conformed to the Declaration of Helsinki, and the Ethics Committee of Doshisha University approved the purpose and all procedures of the study (no. 15085). None of the subjects had cardiovascular abnormalities or skin lesions.

### Water immersion protocol
All of the experiments were performed at the same time of the day in a climatic chamber with its ambient temperature and relative humidity maintained at 25.0 ± 0.5 °C and 55 ± 3%, respectively. After entering the climatic chamber, the subject assumed a sitting position on a chair for 10 min (rest stage) and then immersed his right lower leg up to the knee in a 47-L water container for 15 min in $CO_2$-rich water (1150 ppm, CCWI) or tap water (CWI) maintained at 20 °C (Fig. 1). After the 15-min immersion, the subject

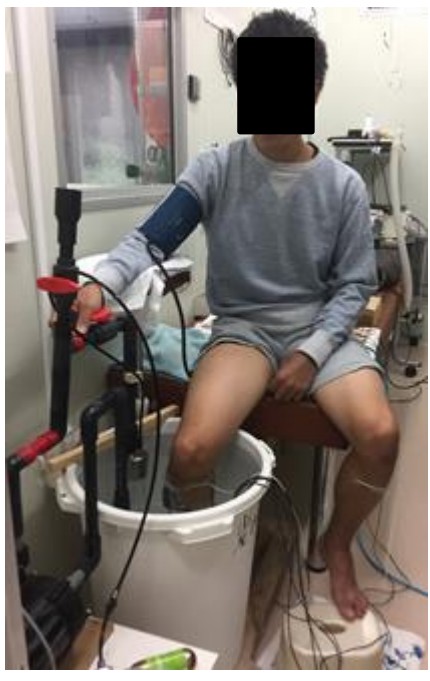

**Figure 1** **Photograph of CO$_2$-rich cold water immresion.** Photograph of a subject immersing the right lower leg up to the knee for 15 min in CO$_2$-rich cold water.

withdrew his lower leg from the CCWI or CWI and then rested for a 20-min recovery period. The subjects performed both a CWI and a CCWI and were not informed whether the water was CCWI or CWI in order to ensure a double-blind design.

The CO$_2$-rich water was prepared by dissolving CO$_2$ in tap water using a dual-chamber/dual-vortex high speed rotation system (DDHRS; type S1, Taikougiken, Kumamoto, Japan). Water was pumped into the DDHRS. Gas was injected into the DDHRS and was reduced in the DDHRS by the centrifugation effect. In this study, the flow rates of CO$_2$ and water were 0.3 L/min and 20 L/min, respectively. This DDHRS can produce a microbubble water flow in which CO$_2$ is dissolved very quickly (within 10 min) by a connection with a small CO$_2$ cartridge (74 g), and the temperature was maintained at 20 °C by an isothermal cooling system (BB310, Yamato, Shiga, Japan).

## Measurements and data analysis

As the subjects' core temperature (T$_{core}$), the sublingual temperature was measured with an electric thermometer (MC-652LC, Omron, Kyoto, Japan). Two skin temperatures were measured at the lower leg (i.e., the tibialis anterior [TA] muscle) under water immersion (T$_{sk}$-WI) and at the subject's non-immersed control leg (T$_{sk}$-cont) with commercial thermistors (LT-8, Gram Co., Saitama, Japan). The thermistors for the lower legs were positioned at the point at one-third of the length of the lateral epicondyle of the femur and the malleolus lateralis line from the lateral epicondyle of the femur and the same length horizontally back.

At the same time as the temperature measurements, the subject's systolic and diastolic blood pressure (SBP, DBP) and heart rate (HR) were recorded by an automated tonometer technique (HBP-1300, Omron) using the subject's upper right arm which rested at heart level alongside his body. The mean arterial pressure (MAP) was automatically calculated with the formula $MAP = DBP + \frac{1}{3}(SBP - DBP)$.

The subject's thermal sensation was reported by the subject every 5 min after the start of the pre-immersion and throughout the experiment. The thermal sensations were rated by the subject based on a conventional seven-point scale (*Fanger, 1984*) comprising cold (−3), cool (−2), slightly cool (−1), neutral (0), slightly warm (1), warm (2), and hot (3). In this study, we limited the thermal sensations to a narrow scale: below −3 to 0.

We measured each subject's skin adipose tissue thickness (ATT) on two regions of the TA and gastrocnemius (GAS) muscles by using an ultrasonic device (SSD-3500SV, Hitachi-Aloka Medical, Tokyo), because the transmissive pulsed light in the NIRS device is influenced by the ATT when the ATT is >5 mm in both of the two muscle regions (*Binzoni et al., 1998*).

NIRS provides a noninvasive window into the microvascular and intramuscular oxygenation status of skeletal muscle under a wide range of conditions at rest (*Adami et al., 2015*; *Binzoni et al., 2000*) and during exercise (*Koga et al., 2013*). In healthy and clinical populations, the total heme concentration (total [Hb+Mb]) and the tissue $O_2$ saturation ($StO_2$) can be estimated (*Yamabata et al., 2016*). In the present study, the oxygenation (oxy[Hb+Mb]) and deoxygenation (deoxy[Hb+Mb]) concentration profiles and their sum (i.e., the total[Hb+Mb] concentration profile) on the two regions of the TA and GAS muscles were simultaneously recorded by two customized continuous-wave NIRS devices (BOM-L1 TR, Omega Wave Co., Tokyo). This system was able to continuously monitor the changes in the oxy[Hb+Mb], deoxy[Hb+Mb], and total[Hb+Mb] concentrations at two different sites (the skin and muscle). Data were collected as previously described in *Yamabata et al. (2016)*. Specifically, we calculated from the light attenuation changes by using a modification of the Beer-Lambert law.

In this study, we defined the 7.5-mm layer as '*superficial*' and the 15-mm layer as '*muscle*'. The TA and GAS muscles with the attached probe holder were then wrapped in a dark-colored elastic bandage to further secure the probes and to eliminate ambient light that might contaminate the NIRS signals. The NIRS data reflected the relative concentration changes in the hemoglobin chromophores and thus did not reflect the absolute tissue $O_2$ values. As we set the probe gain setting at 0 prior to testing the subjects at rest in the sitting position, the relative changes from the resting ($\Delta$oxy[Hb+Mb], $\Delta$deoxy[Hb+Mb], and $\Delta$total[Hb+Mb]) values were calculated. With the use of two detectors at the lower penetration depth of 7.5 mm, the superficial $\Delta$oxy[Hb+Mb], $\Delta$deoxy[Hb+Mb], and $\Delta$total[Hb+Mb] values were assumed to reflect the cutaneous blood flow (i.e., $\Delta$oxy[Hb+Mb]$_{superficial}$, $\Delta$deoxy[Hb+Mb]$_{superficial}$, and $\Delta$total[Hb+Mb]$_{superficial}$).

We also measured the data from the deep penetration depth of 15 mm (i.e., the muscle $\Delta$oxy[Hb+Mb], $\Delta$deoxy[Hb+Mb], and $\Delta$total[Hb+Mb]) values. The $\Delta$oxy[Hb+Mb]$_{muscle}$, $\Delta$deoxy[Hb+Mb]$_{muscle}$, and $\Delta$total[Hb+Mb]$_{muscle}$ were obtained by subtracting these superficial variables from the muscle variables ($\Delta$oxy[Hb+Mb]$_{muscle}$),

$\Delta$deoxy[Hb+Mb]$_{muscle}$, and $\Delta$total[Hb+Mb]$_{muscle}$ according to the modified Beer-Lambert law (*Ando et al., 2013*; *Horiuchi et al., 2016*; *Yamabata et al., 2016*).

From these data, we calculated the $S_tO_2$ as oxy[Hb+Mb]/total[Hb+Mb] $\times$ 100% in the superficial and muscle regions. The NIRS-$S_tO_2$ measurements reflect a mixture of arteriole, capillary, and venous blood flows and do not separate venous from arterial saturations (*Culver et al., 2003*). NIRS-derived $S_tO_2$ values closely correspond to the actual microvascular oxygenation in skeletal muscle ($S_mO_2$) (*Koga et al., 2013*; *Sperandio et al., 2009*; *Wüst et al., 2014*).

We calculated the averaged data of each NIRS parameter for 30 s at each 3-min interval throughout the experiment: water immersion (WI)3 through WI15 at 3, 6, 12, and 15 min, and recovery (rec)3 through rec18 at 3, 6, 12, 15, and 18 min. The baseline values were averaged for the final 30 s of the subject's 10-min rest in a sitting position. The data of the cardiovascular indexes were calculated at each 5-min interval, i.e., baseline; 10-min sitting position; the rest periods, WI for 15 min (WI5, WI10, and W15), and recovery for 20 min (rec5, rec10, rec15, and rec20), averaging the values for 30 s at each period.

### Statistical analysis

All values are presented as the mean $\pm$ standard error (SE). The significance of differences in each variable ($\Delta$oxy[Hb+Mb], $\Delta$deoxy[Hb+Mb], and $\Delta$total[Hb+Mb]) was determined by a two-way analysis of variance (ANOVA) comparing water conditions (CCWI and CWI) $\times$ time periods (rest, WI3–WI15, and recovery at rec3–rec18). A post hoc comparison was applied by Bonferroni test for the appropriate data sets when a significant $F$-value was obtained. A partial eta-square ($\eta^2$) was also determined. Another post hoc comparison was applied by Dunnett's test to determine multiple comparisons from baseline. All analyses were performed using SPSS software (Abacus Concepts, Berkeley, CA) with significance in all cases set at the 5% level.

Regarding the sample determination using PS Power and Sample Size Calculations (*Dupont & Plummer, 1998*), we are planning a study of a continuous response variable from matched pairs of 15 healthy young subjects. Prior data of $\Delta$StO$_2$ of TA and GAS muscles indicate that the difference in the response of matched pairs is normally distributed with the standard deviation 3.486. If the true difference in the mean response of matched pairs is 3.9, we will need to study 8 pairs of subjects to be able to reject the null hypothesis that this response difference is zero with the probability (power) 0.8. The Type I error probability associated with this test of the null hypothesis is 0.05. Therefore, judging from the criteria of the power 0.8, we recognized that the sample size of our 15 pairs of subjects provides the power 0.941.

## RESULTS

### The ATT findings

The average ATT value in the TA region was 3.0 $\pm$ 0.2 mm, and that in the GAS region was 3.9 $\pm$ 0.3 mm.

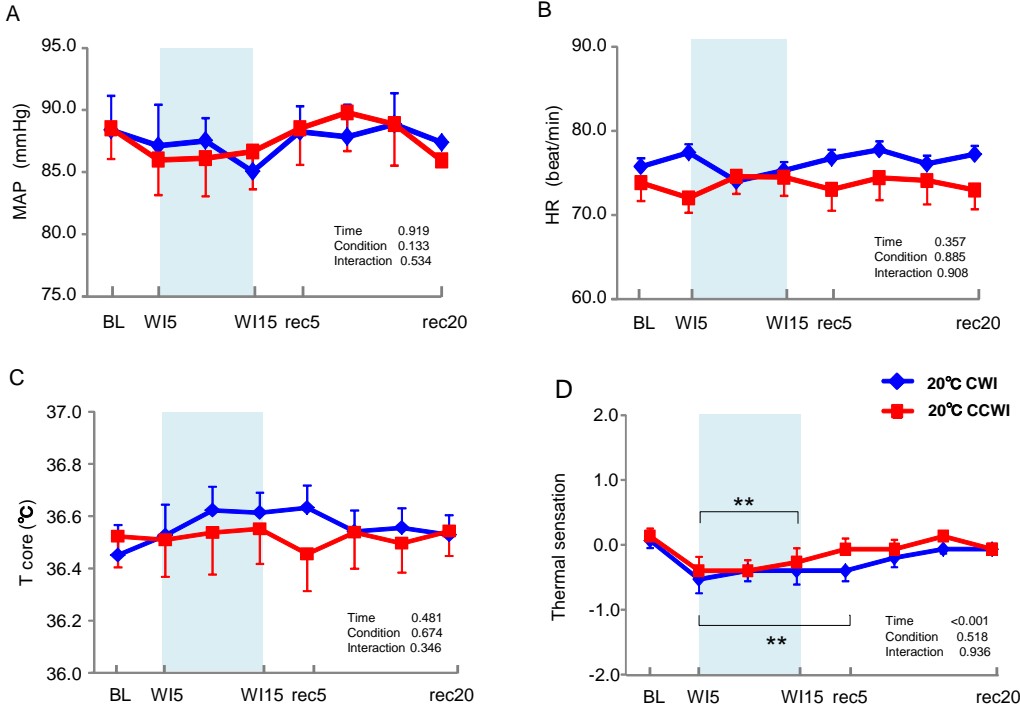

**Figure 2** The subjects' mean arterial pressure, heart rate, core temperature, and visual anagoge scale. The subjects' mean arterial pressure (MAP: A), heart rate (HR: B), core temperature (Tcore: C), and visual anagoge scale (VAS: D) values in the CCWI and CWI protocols. Mean values in 20 sec and standard error are shown throughout the experiment. *Blue circles:* CWI at 20 °C. *Red circles:* CCWI at 20 °C.

## Cardiovascular responses during water immersion

The 15 subjects' MAP values remained unchanged during the immersion and recovery periods, with no significant difference between CCWI and CWI (interaction effect: $F(9,126)$ $= 0.892, p = 0.534$) (Fig. 2A). As shown in Fig. 2B, the HR values during the CCWI remained constant throughout the experiment ($74 \pm 2.2$ beats $\mathrm{min}^{-1}$ at baseline and WI15; $73 \pm 2.3$ beats $\mathrm{min}^{-1}$ at rec20) (interaction effect: $F(9,126) = 0.446, p = 0.908$), which supported the idea that the diving reflex in HR could not be induced with only one leg immersed in the water. These results indicate that both the cold $CO_2$-rich water and cold tap water immersions themselves did not influence the subjects' cardiovascular responses during one-leg water immersion, irrespective of the $CO_2$ condition.

## Core temperature and skin temperatures in the WI and non-WI legs during $CO_2$-rich immersion and recovery

The sublingual temperature ($T_{core}$) values during the CCWI remained constant at a narrow range between 36.4° and 36.6 °C and were not significantly lower than those during the CWI (Fig. 2C) (interaction effect: $F(9,126) = 1.131, p = 0.346$). The $T_{sk}$-WI values decreased abruptly during each water immersion (CCWI: $21.8 \pm 0.1$ °C, CWI: $21.8 \pm 0.2$ °C, at WI15) and returned at the end of the recovery, i.e., rec18 (CCWI: $25.8 \pm 0.2$ °C, CWI: $26.4 \pm 0.2$ °C) (Fig. 3A).

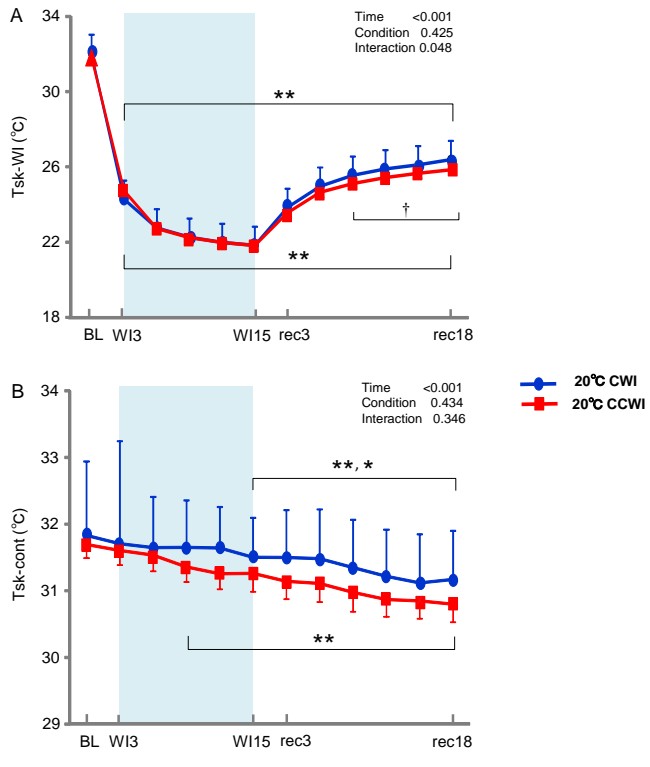

**Figure 3** **The subjects' skin temperature at the immersed leg and non-immersed leg.** Mean values of skin temperature at the immersed leg ($T_{sk}$-WI; A) and non-immersed leg ($T_{sk}$-cont; B). *Blue circles:* CWI at 20 °C. *Red circles:* CCWI at 20 °C. Time effect from baseline: $*p < 0.05$, $**p < 0.01$ from BL. Simple main effect: $†p < 0.05$, $††p < 0.01$ between CCWI and CWI at the same time point.

Compared to the baseline, there was significant lower $T_{sk}$-WI during the CCWI and CWI until the end of recovery (time effect: $F(1,154) = 446.973$, $p < 0.001$, $\eta^2 = 0.970$). Compared to the CWI, the significantly lower $T_{sk}$-WI values continued from rec9 to rec18 of the recovery period following the CCWI (interaction effect: $F(11,154) = 1.901$, $p = 0.048$, $\eta^2 = 0.160$). In addition, the $T_{sk}$-cont values gradually and significantly decreased from baseline to rec18 and were significantly lower from WI9 during the CCWI and from WI15 during the CWI to rec18 (time effect: $F(1,154) = 19.445$, $p < 0.001$, $\eta^2 = 0.599$, Fig. 3B).

It is very noteworthy that the control leg ($T_{sk}$-cont) values tended to be lower during the CCWI than during the CWI until the end of recovery, corresponding to the reduction in $T_{sk}$-WI. In addition, the thermal sensation was significantly decreased during the immersion and began increasing again during the recovery period (Fig. 2D) (time effect: $F(1,126) = 8.985$, $p < 0.001$, $\eta^2 = 0.391$), but it was slightly lower at a later recovery point in the CCWI compared to the CWI (condition effect: $F(1,14) = 0.440$, $p = 0.518$).

## TA muscle

The $\Delta$oxy[Hb+Mb]$_{superficial}$ increased significantly from baseline (by approx. 0.2 AU) during the CCWI (periods WI3 to rec3, $p < 0.01$) and the CWI (periods WI9 to WI12, $p < 0.01$), and this response returned to baseline following the water immersion (time

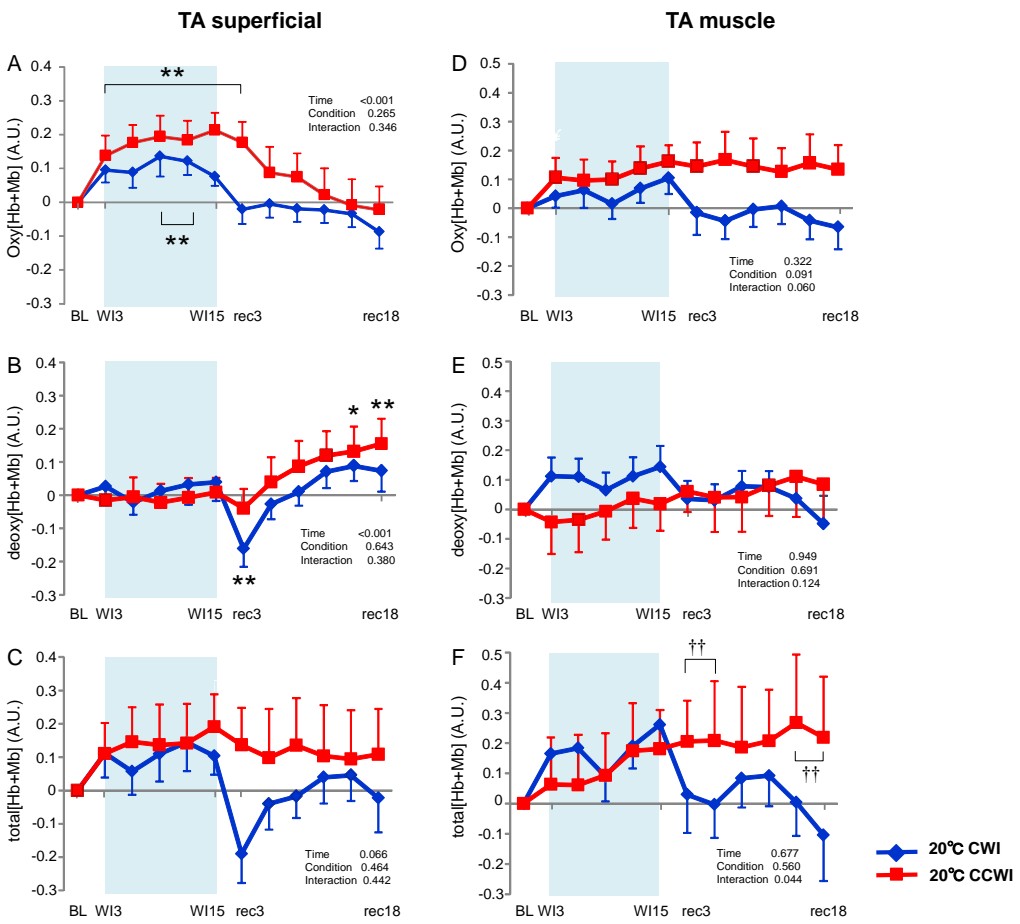

**Figure 4 Changes of oxygenation, deoxygenation, and total hemoglobin kinetics in the tibialis anterior muscle.** Changes of oxygenation (oxy [Hb+Mb]), deoxygenation (Deoxy [Hb+Mb]), and total hemoglobin (total [Hb+Mb]) kinetics from the basal level in the superficial layer (A–C) and muscle layer (D–F) of the tibialis anterior (TA) muscle. Time effect from baseline: $^*p < 0.05$, $^{**}p < 0.01$ from BL. Simple main effect: $\dagger p < 0.05$, $\dagger\dagger p < 0.01$ between CCWI and CWI at the same time point. Time bins and symbols are the same as in Fig. 3.

effect: F(11, 143) = 14.076, $p < 0.001$, $\eta^2 = 0.573$, Fig. 4A). The $\Delta$deoxy[Hb+Mb]$_{superficial}$ during the CCWI remained constant until the end of the water immersion, and it was significantly increased at rec15 and rec18 from baseline due to vasodilation in venous veins during recovery (time effect: F(11, 143) = 5.087, $p < 0.001$, $\eta^2 = 0.298$, Fig. 4B). The $\Delta$total[Hb+Mb]$_{superficial}$ increased by approx. 0.2 AU during water immersion and stayed higher during the recovery following immersion (time effect: F(11, 143) = 1.763, $p = 0.066$, Fig. 4C). Overall, the increase in $\Delta$total[Hb+Mb]$_{superficial}$ was a result of the $\Delta$oxy[Hb+Mb]$_{superficial}$ increase during water immersion, and during recovery this response was a result of the $\Delta$deoxy[Hb+Mb]$_{superficial}$ increase.

The $\Delta$oxy[Hb+Mb]$_{muscle}$ increased gradually from baseline and stayed at a higher level during the recovery period with a moderate ES at all points between CCWI and CWI (ES: 0.67–0.80, interaction effect: F(11, 143) = 1.813, $r = 0.060$, Fig. 4D). By contrast, the

$\Delta$deoxy$[Hb+Mb]_{muscle}$ was unaltered from baseline and increased slightly after the CCWI (Fig. 4E). The $\Delta$total$[Hb+Mb]_{muscle}$ consequently and gradually increased until the end of recovery without a significant increase from baseline, and this trend contributed mostly to the $\Delta$oxy$[Hb+Mb]_{muscle}$ response with a significant difference at some time points during the CCWI compared to the CWI (interaction effect: $F(11, 143) = 1.983$, $p = 0.044$, $\eta^2 = 0.161$, Fig. 4F).

### GAS muscle

In the CCWI protocol, the $\Delta$oxy$[Hb+Mb]_{superficial}$ was increased from baseline and gradually decreased to below the baseline after immersion (Fig. 5A). Notably, the $\Delta$oxy$[Hb+Mb]_{superficial}$ was significantly elevated from baseline (time effect: $F(11, 143) = 9.258$, $p < 0.001$, $\eta^2 = 0.416$). By contrast, the $\Delta$deoxy$[Hb+Mb]_{superficial}$ during the water immersion abruptly decreased to a significantly lower level from baseline in both the CCWI and CWI (time effect: $F(11, 143) = 24.118$, $p < 0.001$, $\eta^2 = 0.650$) and increased again during recovery (interaction effect: $F(11,143) = 2.107$, $p = 0.023$, $\eta^2 = 0.139$, Fig. 5B). During the CCWI, the $\Delta$total$[Hb+Mb]_{superficial}$ tended to be lower than the baseline values, but this decrease was sustained below baseline until the end of recovery (Fig. 5C).

At the GAS muscle region, the $\Delta$oxy$[Hb+Mb]_{muscle}$ was constant throughout the CCWI. After the CWI, $\Delta$oxy$[Hb+Mb]_{muscle}$ gradually decreased until the end of recovery (time effect: $F(11, 143) = 6.602$, $p < 0.001$, $\eta^2 = 0.355$, Fig. 5D). The $\Delta$deoxy$[Hb+Mb]_{muscle}$ was gradually augmented until the end of recovery (time effect: $F(11, 143) = 8.875$, $p < 0.001$, $\eta^2 = 0.425$, Fig. 4E), which reflected mostly the $\Delta$total$[Hb+Mb]_{muscle}$ response (Fig. 5F). By contrast, the $\Delta$total$[Hb+Mb]_{muscle}$ showed the following: a decrease of $\Delta$oxy$[Hb+Mb]_{muscle}$ and an increase of $\Delta$deoxy$[Hb+Mb]_{muscle}$ that offset each other.

### Alternations in $O_2$ saturation inmuscle as measured by NIRS

The $StO_2$ values gradually increased during the water immersion and then returned to a decrease during the recovery period (Supplemental Information). At the 20-min recovery, the $StO_2$ values had fallen to below the baseline. Since the baseline values of $StO_2$ differed between the CCWI and CWI, we estimated the change in the $StO_2$ from baseline ($\Delta StO_2$) to compare the values between the CCWI and CWI and between the TA and GAS muscles (Fig. 6). For the TA muscle, a significant increase from baseline in $\Delta StO_{2muscle}$ reached $7.5 \pm 1.1\%$ at WI9 in the CCWI (time effect: $F(11,143) = 17.025$, $p < 0.001$, $\eta^2 = 0.654$), which was very similar to the $\Delta StO_{2superfical}$ value $8.0 \pm 0.9\%$. It is therefore likely that the $CO_2$ diffusion arrives at deep muscle tissue in the TA. By contrast, for the GAS muscle, the $\Delta StO_{2muscle}$ was increased at $3.6 \pm 1.5\%$ at WI9, which is approx. one-half compared to the $\Delta StO_{2superfical}$ of $7.5 \pm 0.9\%$ at WI9; this suggests that it might be difficult for $CO_2$ to diffuse into the GAS. Apparently, $CO_2$ induced a greater $\Delta StO_2$ in both superficial and muscle regions in both the TA and GAS (interaction effect: $p = 0.044-0.002$, $\eta^2 = 0.61-0.251$).

## DISCUSSION

Using NIRS, we examined two common assumptions regarding the application of $CO_2$-rich cold water immersion (CCWI) to (1) the changes in skin and muscle blood perfusion

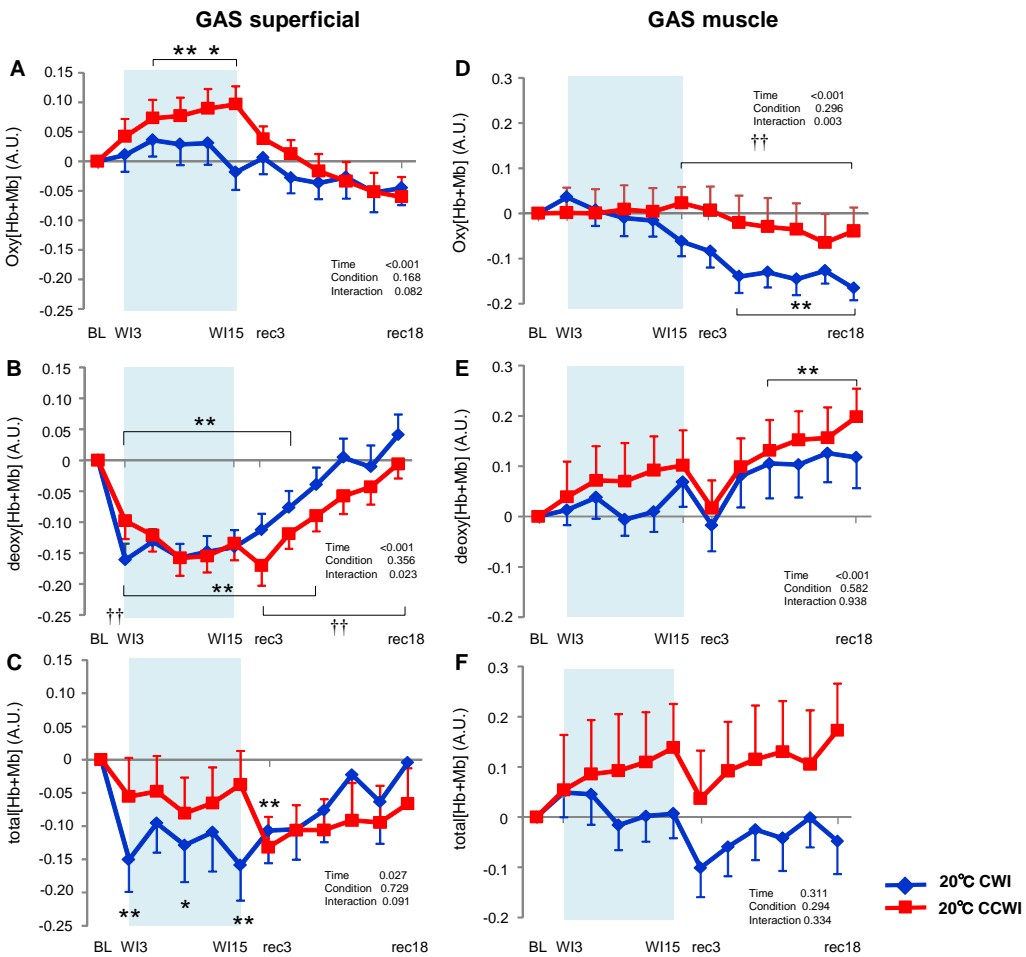

**Figure 5 Changes of oxygenation, deoxygenation, and total hemoglobin (total [Hb+Mb]) kinetics in the gastrocnemius (GAS) muscle.** Changes of oxygenation (oxy [Hb+Mb]), deoxygenation (Deoxy [Hb+Mb]), and total hemoglobin (total [Hb+Mb]) kinetics from the basal level in the superficial layer (A–C) and muscle layer (D–F) of the gastrocnemius (GAS) muscle. Time effect from baseline: $*p < 0.05$, $**p < 0.01$ from BL. Simple main effect: $†p < 0.05$, $††p < 0.01$ between CCWI and CWI at the same time point. Time bins and symbols are the same as in Fig. 3.

related to capillary-venous heme concentration and the volume of small vessels, and (2) the promotion of the heat transfer from the body to the water due to vasodilation during CCWI. To examine these assumptions, we first used a resting cold water immersion protocol to determine the agreement between $CO_2$-induced vasodilation and the change in leg skin temperature. Our results indicate that during the recovery period after water immersion, the heat dissipation could be promoted by using CCWI rather than CWI, because a significantly lower $T_{sk}$-WI continued until the end of the recovery period. The NIRS profiles of the oxy[Hb+Mb]$_{superficial}$ responses in both the TA and GAS muscles may also support the concept that CCWI in greater increases of cutaneous blood flow and blood volume with different $\Delta$deoxy[Hb+Mb]$_{superficial}$ values.

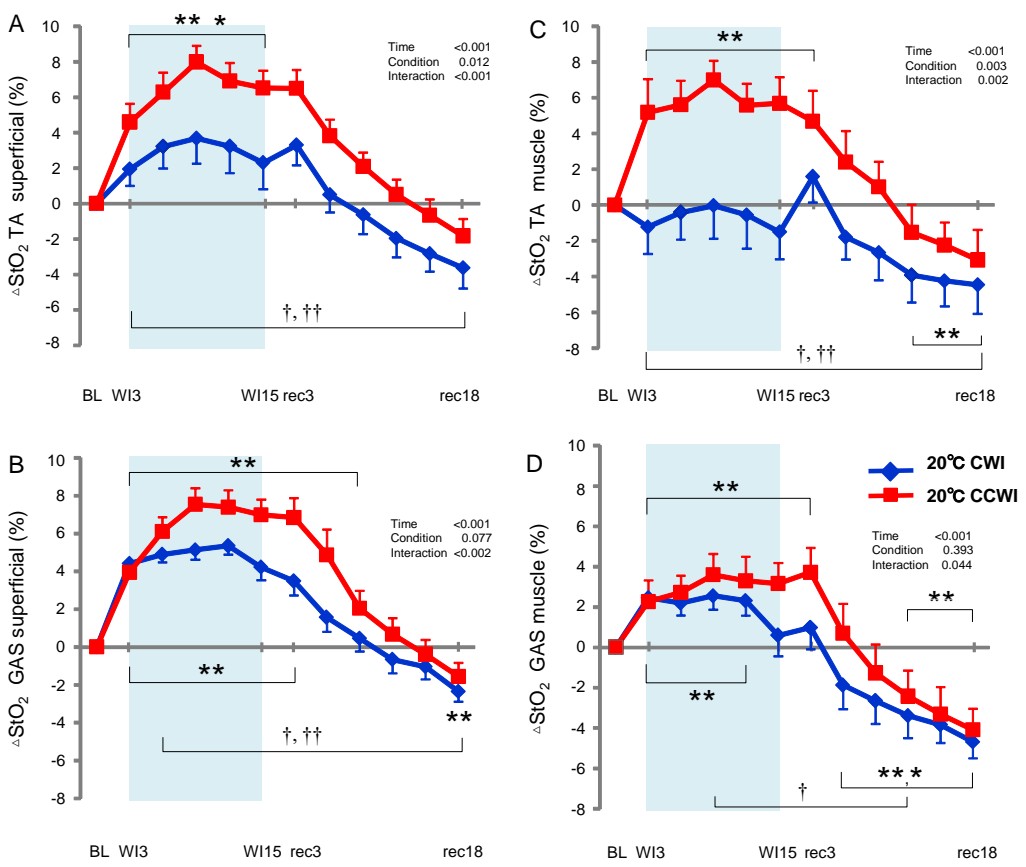

**Figure 6  Changes of muscle oxygen saturation ($\Delta StO_2$) in TA muscle and GAS muscle between the CCWI and CWI.** Changes of muscle oxygen saturation ($\Delta StO_2$) in TA muscle (A, C) and GAS muscle (B, D) between the CCWI and CWI. Time effect from baseline: $*p < 0.05$, $**p < 0.01$ from BL. Simple main effect: $†p < 0.05$, $††p < 0.01$ between CCWI and CWI at the same time point. Time bins and symbols are the same as in Fig. 3.

In addition, we assessed whether $\Delta deoxy[Hb+Mb]_{muscle}$ was not affected by the changes in $\Delta total[Hb+Mb]_{muscle}$. Even though the increase in $\Delta deoxy[Hb+Mb]_{muscle}$ at the GAS muscle contributed to the $\Delta total[Hb+Mb]_{muscle}$ during water immersion, we also observed that the increase in $\Delta deoxy[Hb+Mb]_{muscle}$ corresponded to the increase in $\Delta total[Hb+Mb]_{muscle}$ during the recovery period irrespective of different muscles or muscle layers. It is difficult to comprehensively interpret our results in part due to the complexity of three interactive factors of cold stimulus, hydrostatic pressure, and $CO_2$ vasodilation effects. We should thus take into account that two separate periods between water immersion and recovery may be altered by different physiological mechanisms.

## $CO_2$ promotes more the heat transfer from the body to the water and lower skin temperature during CCWI than CWI

Our analyses revealed that compared to the CWI, the CCWI induced a slightly lower $T_{sk}$-WI and a slightly higher thermal sensation at the recovery period, which indicates

that CCWI facilitates not only a decline in $T_{sk}$-WI but also a less cold sensation following water immersion. These observations confirm that an increase in cutaneous blood flow during CCWI may elicit excess heat transfer even if the water temperature is cool. $CO_2$ applied topically can increase the cutaneous blood flow (*Diji, 1959*; Komoto et al., 1986; *Ito, Moore & Koss, 1989*; *Hartmann, Bassenge & Pittler, 1997*) due to cutaneous vasodilation (*Ito, Moore & Koss, 1989*). In the present investigation, this effect promoted not only the heat transfer from the body to the water and lower $T_{sk}$-WI values; it also produced lower $CO_2$-induced $T_{sk}$-cont values, which is one of the strong countermeasures of a hot environment in sports. Moreover, compared to the CWI, the CCWI facilitated the slightly lower cold sensation. It has been demonstrated that $CO_2$ inhibits the activity of cold receptors and facilitates the activity of warm receptors of the skin (*Dodt, 1956*). Such modifications in the activity of skin receptors by $CO_2$ can explain the elevation of the thermal sensations that we observed during the CCWI compared to the CWI, since thermal sensations are caused predominantly by the signals from skin receptors rather than central receptors (*Hensel, 1981*).

We set the water temperature at 20 °C because cold receptors are discharged most vigorously at skin temperatures at 25 °C (*Tansey & Johnson, 2015*) and cold immersion even at 20 °C in this study would have excited a greater number of stimulated cold receptors. In a review of CWI studies, a water temperature at ∼14 °C was most commonly selected with a ≤15-min immersion duration (*Choo et al., 2018a*). We selected 20 °C with a longer immersion duration (15 min) for the present experiment, and the $T_{sk}$-WI was reduced from 31.8 ± 0.2 °C to 21.8 ± 0.1 °C at the end of the immersion, indicating that the underlying muscles were probably cooled to a significant degree and to the same extent between CCWI and CWI (*Ihsan et al., 2013*).

We expected that in the CCWI, increased cutaneous blood flow due to cutaneous vasodilation via $CO_2$ would facilitate the heat transfer from the subject's body to the water. Indeed, an increased $\Delta$oxy[Hb+Mb]$_{superficial}$ was observed even during the CCWI. Consequently, the $\Delta$total[Hb+Mb]$_{superficial}$ at the TA muscle tended to be higher than the baseline (time effect: $p = 0.066$), and fell significantly from the baseline in the GAS muscle (time effect: $p = 0.027$). Our NIRS results thus support our hypothesis that CCWI depresses the marked reduction in the cutaneous blood flow.

## The impact of hydrostatic pressure during water immersion leads to inflow into muscle

Hydrostatic pressure might reduce edema and inflammation by increasing the pressure gradient between the interstitial and intravascular spaces, promoting the re-absorption of interstitial fluid in a manner similar to compression stockings (*Partsch, Winiger & Lun, 2004*). The combination of a cold stimulus and hydrostatic pressure could also act synergistically; decreased muscle temperature may reduce edema by reducing the muscle perfusion (cold-induced vasocontraction) and fluid diffusion into the interstitial space (*Yanagisawa & Fukubayashi, 2010*; *Gregson et al., 2011*), as well as through a reduced permeability of cellular, lymphatic, and capillary vessels (*Coté et al., 1988*)). This might complement any hydrostatic pressure effects on interstitial–intravascular fluid movement.

In the present study, the $\Delta$oxy[Hb+Mb]$_{muscle}$ gradually increased from baseline and stayed at a relatively higher level during recovery. Overall, the increase in $\Delta$total[Hb+Mb]$_{muscle}$ in the TA muscle was a result of the $\Delta$oxy[Hb+Mb]$_{muscle}$ increase during the CCWI, and this response was a result of both the $\Delta$oxy[Hb+Mb]$_{muscle}$ and $\Delta$deoxy[Hb+Mb]$_{muscle}$ during the recovery period. *Choo et al. (2018b)* reported that the skin blood flow had notably less influence on the total[Hb+Mb] signal during the recovery after cooling compared to a non-cooling recovery. However, our findings regarding the $\Delta$total[Hb+Mb]$_{muscle}$ kinetics differed, and they indicate that $CO_2$ could influence the muscle blood flow increase.

By contrast, in the GAS muscle in the CCWI, the $\Delta$oxy[Hb+Mb]$_{muscle}$ remained unchanged throughout the experiment. The gradually augmented $\Delta$deoxy[Hb+Mb]$_{muscle}$ values were mostly reflected by the $\Delta$total[Hb+Mb]$_{muscle}$ response. The results of several intervention studies suggest that the deoxy[Hb+Mb]$_{muscle}$ profile is likely to have different effects on the $O_2$ extraction-related changes and volume-related changes (*Adami et al., 2015*; *Binzoni et al., 2000*; *Truijen et al., 2012*). Thus, CCWI intervention without shivering would improve the redistribution of blood flow from the non-immersion site to the immersion site. If the 20 °C cold exposure reduces the metabolic demand, an abrupt decline in the $\Delta$deoxy[Hb+Mb]$_{muscle}$ from baseline would occur; however, the above result indicated no decline of the $\Delta$deoxy[Hb+Mb]$_{muscle}$ in the TA or GAS muscles.

### Different oxygenation in the TA and GAS regions during water immersion

We observed significant increases in the $\Delta$StO$_2$ in both the TA and GAS muscles during the CCWI (time effect: $p < 0.001$). Apparently, $CO_2$ induced a greater $\Delta$StO$_2$ in both superficial and muscle regions in both the TA and GAS. However, for the TA muscle, there was a similar increase from the baseline in the $\Delta$StO$_2$ between the skin layer and muscle layer during the CCWI. By contrast, for the GAS muscle, the $\Delta$StO$_{2muscle}$ was increased by one-half compared to the $\Delta$StO$_{2superfical}$, suggesting that it might be difficult for $CO_2$ to diffuse into the GAS. It may be possible that specific characteristics of TA muscle account for muscle fiber-type-dependent muscle oxygenation responses to a combined cold and $CO_2$ stimulus, because >70% of the TA muscle in humans consists of only slow-twitch fibers (*Dahmane et al., 2005*). Slow-twitch fibers have more developed muscle capillaries than fast-twitch fibers, which means that the vasodilatory effect of $CO_2$ is further improved.

### Underlying mechanism of vasodilation during $CO_2$ water immersion

A plausible mechanism of $CO_2$-induced vasodilation is associated with extracellular acidosis. Traditional studies demonstrated that acidosis might reduce the contractility of the vascular smooth muscle, leading to vasodilation (*Tobian, Martin & Eilers, 1959*; *Vanhoutte & Clement, 1968*). The reduction in smooth muscle contractility has been ascribed to a reduction in calcium influx or to the suppression of myofilament contractility (*Breemen et al., 1972*; *Fabiato & Fabiato, 1978*). An in vitro study examining the contractility of the rat aorta exposed to a small change in pH (from 7.4 to 7.0) demonstrated that even this small change in pH could reduce vascular smooth muscle

contractility (*Loutzenhiser et al., 1990*). That study also indicated that $H^+$-induced vasodilation is associated with an increase in the amount of calcium sequestered in the norepinephrine-sensitive intracellular calcium store.

Other investigations of the coronary, cerebral and aortic circulations have shown that nitric oxide (*Fukuda et al., 1990*; *Gurevicius et al., 1995*; *Aalkjær & Peng, 1997*) and the activation of potassium channels (*Ishizaka & Kuo, 1996*) may contribute to this acidosis-induced vasodilation. Both skin and muscle arterioles are known to be regulated by sympathetic outflow and vasodilatory substances such as nitric oxide (*Hickner et al., 1997*; *Kellogg Jr, 2006*; *Bernjak et al., 2012*). However, we did not measure the blood substances of pH, potassium, or nitric oxide or their underlying mechanisms in this study.

### Study limitations

To avoid the shivering with which an involuntary muscle contraction of immersed muscles can increase metabolic demand (*Proulx, Ducharme & Kenny, 2003*) and the associated alteration of NIRS profiles, we set the water temperature of CWI and CCWI at 20 °C, a temperature at which subjects can easily tolerate the cooling muscle treatment. Although we did not directly measure our subjects' shivering response, there is only a slight possibility that shivering occurred in light of the subjects' thermal sensations.

Compared to the more global information provided by blood flow in a large supplying artery, the use of NIRS as a monitoring technique provides the ability to assess the specific effects on the tissue of interest. NIRS cannot discriminate between myoglobin (Mb) and hemoglobin (Hb) saturation or between arterial and venous compartments; thus, no unambiguous explanation can be provided. Irrespective of the underlying reasons, the increase in tissue oxygenation that we observed herein was a very consistent feature of the hemodynamic response to the compression of resting muscle.

### Athletic application of $CO_2$-rich water immersion

Cold-water immersion is one of the beneficial recovery techniques that is commonly used by athletes post-exercise to promote the restoration of body systems to baseline conditions and return the physiological system to a pre-exercise state (*Bleakley & Davison, 2010*). Our present experiment was conducted with the subjects in a resting condition without a post-exercise state, and thus the initial muscle blood flow at the onset of water immersion was quite different from that in the above-cited study of athletes. We thus conducted a second experiment to explore whether this CCWI protocol at post-exercise is an effective intervention to maintain exercise performance, which is associated with a reduced blood lactate concentration and reduced heart rate (Supplemental File). Further research regarding the changes in oxygenation profiles following athletic exercise is of interest.

## CONCLUSION

Cold-water immersion with a rich $CO_2$ concentration (CCWI) induced greater vasodilation in both the gastrocnemius and tibialis anterior muscles, which represented apparent increased $\Delta$oxy[Hb+Mb]$_{muscle}$ during CCWI and/or $\Delta$deoxy[Hb+Mb]$_{muscle}$ during the

recovery period. Cold water at 20 °C does not dampen the local metabolic demand during CCWI. It is clear from the fact that $\Delta$deoxy$[Hb+Mb]_{muscle}$ is not reduced. Therefore, a significantly increased $\Delta StO_2$ during CCWI represented rich arterialized $O_2$ content in the muscle tissue compared to CWI. Our results also demonstrated that the lower $T_{sk}$ at the immersed lower extremity was associated with vasodilation by $CO_2$. The use of CCWI could continue to promote the heat transfer from the body to the water after CCWI for $\leq$20 min.

### Funding

This study was supported by the grants from the Japan Science and Technology Agency (JST) Regional Industry-Academia Value Program (VP29117939241, to Yoshiyuki Fukuoka). The funders had no role in study design, data collection and analysis, decision to publish, or preparation of the manuscript.

### Grant Disclosures

The following grant information was disclosed by the authors:
Japan Science and Technology Agency (JST) Regional Industry-Academia Value Program: VP29117939241.

### Competing Interests

Mhiho Yoshimura was employed by Mizuno Corporation after graduation from Doshisha University. As an academic adviser, Hiroaki Tsutsumi is still associated with Taikohgiken Ltd. company.

### Author Contributions

- Miho Yoshimura and Yoshiyuki Fukuoka conceived and designed the experiments, performed the experiments, analyzed the data, prepared figures and/or tables, authored or reviewed drafts of the paper, and approved the final draft.
- Tatsuya Hojo conceived and designed the experiments, performed the experiments, prepared figures and/or tables, authored or reviewed drafts of the paper, and approved the final draft.
- Hayato Yamamoto and Misato Tachibana performed the experiments, analyzed the data, prepared figures and/or tables, and approved the final draft.
- Masatoshi Nakamura performed the experiments, analyzed the data, authored or reviewed drafts of the paper, and approved the final draft.
- Hiroaki Tsutsumi performed the experiments, authored or reviewed drafts of the paper, and approved the final draft.

### Human Ethics

The following information was supplied relating to ethical approvals (i.e., approving body and any reference numbers):
Ethical approval for this study was obtained from Doshisha University (No.15086).
## Ethics

The following information was supplied relating to ethical approvals (i.e., approving body and any reference numbers):

First, written informed consent was obtained from all participants after a detailed explanation about all procedures, the purpose of the study, and the possible risks and benefits of the participation. This study conformed to the Declaration of Helsinki, and the institutional ethical committee of Doshisha University approved the purpose and all procedures of the study (no. 15085).

## Data Availability

The raw measurements and table are provided in the File S1.

## Supplemental Information

Supplemental information for this article can be found online at http://dx.doi.org/10.7717/peerj.9785#supplemental-information.

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
