# Peer review of "Application of carbon dioxide to the skin and muscle oxygenation of human lower-limb muscle sites during cold water immersion"

_PeerJ, doi:10.7717/peerj.9785_

## Round 0.1 · original submission · Major Revisions

The two reviewers and I are impressed with many aspects of the research study and the way it is presented in this manuscript. However, we feel there are still many ways in which this manuscript can be improved. Please make a concerted effort to attend to each of the reviewer's comments in your revised submission.

Reviewer 1 ·

Basic reporting

The authors prepared a very interesting study, focused on a useful recovery method. In my opinion, this research is useful in the training recovery programs and the main findings should be used in athletes' recovery.

In general, the English language is good, while some parts of the Introduction and discussion should be re-written, in terms of some language errors.

The literature references were prepared correctly, while I suggested adding a few other references to the Introduction and Discussion. In general, the authors presented the main findings from previous studies.

The article structure, including figures and table, is correct. The figures are very clear.

The hypothesis should be clarified, but the main hypothesis is consistent with the results.

Experimental design

This study is original research, with an interesting research design. In my opinion, an additional section should be added - including study design. This would increase the value of this paper. For sure, it would be more clear for the reader.

I have marked in the comments for the authors - the aim of the study should be more specifically included in the introduction.

Ethical standards are included. The authors described the approval and the Helsinki Declaration.

The methods and measurements are very specifically described.

Validity of the findings

The research protocol is described as correct, but the research design should be more precise. The study is very interesting and fills the lack of knowledge in recovery and water immersion.

The results are very well presented and described. The figures are clear, without any errors.

Conclusions are understandable but should be re-written, including the main findings and main aim of this study.

Additional comments

Major comments:

Introduction:
First of all, the introduction should be written more logical. In this current version it is not very clear. Please re-arrange this section according to my comments and suggestions.
1. CWI – review of previous paper, including information about different protocols – temperatures and duration, and physiological mechanism – a clear statement.
2. Carbon dioxide (CO2) – What are the main outcomes of apply CO2 in cold water therapy?
3. Connect CCWI with near-infrared spectroscopy – especially in the first part of this paragraph, the reader may found lack of information.
4. Explain, why CCWI is important in recovery.
5. Hypothesis and clear goal of the study.

Line 56-64: In my opinion, the first paragraph should include a smooth description of cold water therapy and recovery. Please provide more information according to physiological mechanisms following CWI. Please provide publication review according to different possibilities to apply CWI in recovery. Please check papers written by Šrámek et al. (2000), Bleakley et al. (2010) and Ihsan (2016).

Šrámek, P., Šimečková, M., Janský, L., Šavlíková, J., & Vybíral, S. (2000). Human physiological responses to immersion into water of different temperatures. European journal of applied physiology, 81(5), 436-442.
Bleakley, C. M., & Davison, G. W. (2010). What is the biochemical and physiological rationale for using cold-water immersion in sports recovery? A systematic review. British journal of sports medicine, 44(3), 179-187.
Ihsan, M., Watson, G., & Abbiss, C. R. (2016). What are the physiological mechanisms for post-exercise cold water immersion in the recovery from prolonged endurance and intermittent exercise?. Sports Medicine, 46(8), 1095-1109.

Line 75: Please provide a statement connecting CWI and near-infrared spectroscopy. This two paragraphs should be connected and logical.

Line 73-92: It is not known, what is the reason to use CCWI? Please explain in the last part of the introduction. Please add also about different temperatures for CWI. It should be shown a analysis of prior studies.

Materials and methods:
1. Have you calculated sample size and power for this study? It would be worth to add, what is the minimal study group needed.
2. Were the participants physically active?
3. Have you considered included and excluded criteria?
4. What is the reason to prepare this research - CWI and CCWI with temperature 20oC?

Line 117-118: “The subjects were not informed whether the water was CCWI or CWI.” Please provide in sub-section - Participants' characteristics or a special “study design”. It would be worth to explain the study design and the main metrological aspects.

Discussion
Line 316-317: “Using NIRS, this study examined two common assumptions in the application of CO2 rich cold water immersion to change in skin- and muscles-blood perfusion related to hematocrit,” Could you please explain this statement. I have not found any information about relation with hematocrit before, in the introduction. Please explain and clearly this.

Please discuss the effect of CWI using protocols with different temperatures. In previous papers you may find few reviews, e.g. Bleakley and Davison, 2010.

Study limitations:
Please indicate point by point limitations, with brief explanation. In my opinion this section should be re-arranged and clarified.
None post-exercise state should be marked as the major limitation with explanation.

Athletic application of CO2-rich water immersion
Please indicate more specifically practical implications.

Conclusions
In my opinion, the conclusions should be re-write to a more consistent form.



Minor comments:

Abstract:
Line 28-29: “Cold therapy has a disadvantage in that a cold stimulus can induce vascular constriction in arterial and venous capillaries”. Please re-arrange this sentence.

Introduction:
Line 56-57: “Cold therapy has a disadvantage in that a cold stimulus can induce vascular constriction in arterial and venous capillaries”. Please re-arrange this sentence.

Line 57-59: “As a result, the decreased blood flow in both peripheral and deep vasculature would lead to a delayed accumulation of substances that are related to muscle damage and muscle fatigue after competitive sports events.” Please consider to move this sentence at the end of the paragraph.

Line 65-67: “Over the last three decades, researchers observed that the effects of carbon dioxide (CO2) hot water depend mainly on not only the cutaneous vasodilation but also the muscle vasodilation elicited by the CO2 that diffuses into the subcutaneous tissues through the skin layers”. Please re-arrange this sentence. Grammatically should be improved.

Line 72-74: “We hypothesized that the arterial and venous vasodilation in both superficial and inside-muscle layers of different muscles would be significantly augmented by enriched CO2, even though during CWI the cold stimulus induces vascular constriction.” This sentence should be moved to the end on the introduction. Try to connect with the aim of the study.

Materials and methods:

Line 114, 116: Please do not begin sentence with – “He …” “his”

Line 127-137: Move to the Introduction

Statistical analysis:

I would recommend adding eta square to ANOVA analysis.

Conclusions:
Line 515: Please change term „leg” to “lower extremity”.

Reviewer 2 ·

Basic reporting

The manuscript has numerous grammatical and structural errors and requires proof reading:
- Some sentences can be written in a more concise manner. For example, Line 29 Abstract, can be revised to “…CO2 hot water immersion results in vasodilation in both cutaneous and muscle tissues”. Line 73 – “vascular constriction” can be replaced with “vasoconstriction”.
- Define the abbreviation for the spelled-out text at the first mention. For example, skin temperature was mentioned first in Line 32 (Abstract), but the abbreviation “Tsk” was defined in Line 37.
- Some abbreviations were redefined throughout the manuscript. The authors can consider using the abbreviations after they have been defined at the first mention. For example, Tcore was defined in Line 136, Line 239, and Tsk-cont was defined in Line 139 and Line 249.
- There was some inconsistency when referring to a parameter, for example, thermal sensation versus thermal score in the figure and VAS in the figure caption.
- Replace “became significantly increase” with “increased significantly”.
- Some inconsistency in the format of the abbreviations, for example, core and sk were subscripts in the methods section, but not in Line 239 and Line 241.

Table 1 and Figure 6 are presenting the same variable StO2 for TA and GAS (superficial and muscle), one in absolute, the other in relative change. I suggest the author to include Table 1 as supplementary material only.

Figure 4, 5, 6 – “Time bins and symbols are the same as in Fig. 3”. Define the abbreviation and symbols in each figure.

Experimental design

The water immersion protocol is well-described. The authors can consider using a schematic diagram to help readers to better understand the flow of the protocols and the time points of different measurements, especially for the NIRS measurements. It is also important to indicate if the CCWI and CWI conditions were conducted on separate visits, and how the order of conditions was decided.

Validity of the findings

Throughout the results section, there is some inconsistency and some areas require clarification:
- Main and interaction effects, and F-values were not reported for HR, Tcore, although these were reported for other variables.
- Was post-hoc analysis for Tsk-WI, Tsk-cont and thermal sensation performed? As there was a significant time effects, but the exact p values for the mentioned time points were not included.
- Effect size should only be reported if the p-values approximated 0.05. Additionally, p-values should be presented for Line 269.
- Include the p values, for example, ∆total[Hb+Mb] (Line 264), ∆deoxy[Hb+Mb] (Line 269).

Line 331 – Based on the results, there was only a main time effect, but no condition or interaction effect for thermal sensation. Hence the sentences “…a slightly higher thermal score…” an “CCWI facilitated the slightly but significantly lower cold sensation compared to CWI” is not supported by the results.

Line 363 to 411 – A large portion of this section about hydrostatic pressure is informative but has little relevancy. The discussion about the hydrostatic effect is distracting from the main aim of the study, which is to compare between CCWI and CWI, in which hydrostatic pressure is present in both conditions.

Line 377 – Cooling per se may also reduce muscle temperature and metabolic activity, thus resulting in increased oxyHb. Additionally, I recommend the authors to avoid interpreting their results based on effect size only especially when the associated p-value is not close to 0.05.

The changes in the NIRS parameters should be discussed in conjunction with the underlying mechanism of vasodilation during CO2-enriched CWI.
Line 413 to 427 – This paragraph reads like the results section.

Line 418 to 420 – This sentence reads like the authors attributed the difference in the oxyHb between CWI and CCWI to hydrostatic pressure, but the hydrostatic effect is present in both conditions and thus cannot adequately explain the stated difference.

Line 447 to 459 – As only the lower leg is immersed, it is unlikely that HR, MAP and core temperature will be changed drastically. This section has little relevancy to the focus of the study.

The conclusion was not supported by the results. As mentioned in my comments earlier, avoid interpreting the results based on effect size especially when the p-value is not close to 0.05:
- Figure 5 - OxyHb and deoxyHb were not different between CWI and CCWI, although total Hb increased after CCWI in the TA muscle.
- Figure 6 – OxyHb was higher in CCWI, while total Hb and deoxyHb was not different in the GAS muscle.
- Higher StO2 is also indicative of a lower metabolic activity (Ihsan et al., 2013).

Additional comments

This study examines the effect of CO2-enriched CWI on skin and muscle perfusion and oxygenation. In general, the study appears to be carefully performed and well-controlled. They conclude that CO2-riched CWI causes vasodilation and decreases skin temperature to a greater extent compared with CWI. This reviewer finds that the conclusion is not supported by their results. The basis for this study is that CO2-enriched CWI will induce vasodilation during immersion. However, taken together the changes in total Hb (perfusion) for TA and GAS (both superficial and deep tissues), there was no difference between CWI and CCWI, except for total Hb for TA which appeared to increase post-immersion, but post-hoc analysis was not shown (see below). The results are also in contrast to the findings reported by Nishimura et al. (2002) “Effects of repeated carbon dioxide-rich water bathing on core temperature, cutaneous blood flow and thermal sensation”, and Tanaka et al. (2020) - Body cooling effects of immersion of the forearms in high-concentration artificial carbonic acid water at 25°C. In both studies, skin blood flow as assessed by Laser Doppler increased significantly while thermal sensation was lower in the CO2-enriched CWI:

- Total Hb for TA superficial as an indicator of cutaneous perfusion was not different between CWI and CCWI and there was no interaction effect (Figure 4, condition effect p = 0.464, time effect p = 0.066, interaction effect p = 0.442).
- Total Hb for TA muscle as an indicator of muscle perfusion showed that it increased post-immersion in CCWI, as opposed to decrease in CWI (Figure 4, interaction effect p = 0.044, post-hoc analysis not shown).
- Total Hb for GAS superficial also showed no difference between CWI and CCWI (Figure 5, condition effect p = 0.729, time effect p = 0.027, interaction effect p = 0.091).
- Total Hb for GAS muscle showed no effect of CCWI or CWI (Figure 5, condition effect p = 0.311, time effect p = 0.294, interaction effect p = 0.334).

---

## Round 0.2 · Minor Revisions

I wish to thank you for your efforts in attending to many of the reviewers comments. Please attend to the remaining reviewers comments so that we can consider this manuscript for publication in PeerJ.

Reviewer 1 ·

Basic reporting

The authors prepared very carefully all comments and suggestions. All criterias are included correctly.

Experimental design

The experimental design is prepared correct, however, I would suggest adding statistical analysis of the sample size - using G Power. The authors recruited 15 participants. It would increase the quality of this work.

Validity of the findings

This point of view is a novel investigation which may find pracctical application is profesional sport.

Reviewer 2 ·

Basic reporting

Congratulations on a vastly improved manuscript and it clarifies the significance of the study. The flow of the introduction and discussion in the paper are much more clearly related to the aims of the study.

Experimental design

Methods:
Line 128 – Thank you for clarifying the blinding process. Just one more question, a double-blind design means that both the subject and the researcher involved were unaware of the condition (CWI or CCWI) during the experiment. Is that the case? You can also consider removing the parenthesis in that sentence and replace the word “insure” with “ensure”.

Statistical analysis:
Line 209 –Thank you for addressing my concerns regarding the use of effect size. Since the effect sizes are not reported in the revised manuscript, consider removing the related section in here.

Validity of the findings

The results are much clearer and the conclusion is in line with supporting data.

Additional comments

Abstract:

Line 51 - Consider removing “despite the lower Tsk-WI values after the CCWI…” since you stated that Tsk-WI remained lower in the CCWI than in CWI later in the same sentence.

Results:

Line 292 – Replace “Alternation” with “alteration” and “under” with “in” – Alterations in O2 saturation in muscles as measured by NIRS

Discussion:

Line 310 – consider remove “blood vessels” in the sentence blood vessels' vasodilation since vasodilation refers to the widening of blood vessels.

The discussion about thermal sensation is a little hard to follow. Line 330 to 331 states that CCWI results in a higher thermal sensation. If based on the scale used (cold = -3, neutral = 0), a higher thermal sensation means that the subjects felt less cold. But in Line 338, CWI results in lower cold sensation. Also, consider removing “significantly” in the sentence “CCWI facilitated the slightly but significantly lower cold sensation”. The absence of a significant difference in thermal sensation between CCWI and CWI may also be related to relative small segment of the body exposed compared with whole body immersion.

Line 369 to 371 - I am not sure if this sentence is necessary here as it is repetitive with the next paragraph.

Line 373 - This sentence is hard to follow. The change in total Hb (muscle) was a result of oxyHb(muscle) increase during CCWI and recovery, which was a result of oxyHb(muscle) and dexoyHb(muscle)? Consider rewriting for clarity.

Line 380 – Consider removing “our subjects’”.

Line 383 – Consider including more than one reference since you mentioned several studies have suggested…

Line 385 – Please clarify redistribution of blood flow from where to where?

Line 399 – Even though it may be obvious to some readers, it may be better to clarify why slow-twitch fibers would facilitate the diffuse of CO2?

Conclusion:

Line 450 – Because of the choice of modal verbs used (“could not” and “not able to”), the current sentence reads like the aim of CCWI was to reduce Δdeoxy[Hb+Mb]muscle and to decrease local metabolic demand. Consider rewriting the sentence for clarity.

---

## Round 0.3 · accepted · Accept

I thank the authors for their hard work in attending to the reviewers comments. On this basis, I am happy to recommend your paper be accepted for publication in PeerJ.